# Experimental Capture Width Ratio on Unit Module System of Hybrid Wave Energy Converter for Nearshore

**Min-Su Park** [1,*] **, Seung-Heon Lee** [2] **and Sang-Cheol Ko** [3]

1   Department of Structural Engineering Research, Korea Institute of Civil Engineering and Building Technology, Goyang 10223, Korea
2   Energy Plus, Gwangju 61042, Korea; sanha241@gmail.com
3   Graduate School of Carbon Convergence Engineering, Jeonju University, Jeonju 55069, Korea; scko@jj.ac.kr
*   Correspondence: mspark@kict.re.kr

**Abstract:** This study proposes a new hybrid wave energy converter composed of a horizontal cylinder and a swing plate to improve the capture width ratio. The horizontal cylinder generates electrical energy by using the potential energy of the incident wave, whereas the swing plate produces electrical energy by using the kinetic energy of the water particles. The converter can improve the capture width ratio of the wave energy by efficiently combining the energies generated by these two different sources. The power-generating performance of the proposed hybrid wave energy converter is evaluated experimentally through a hydraulic model test at a scale ratio of 0.3 in a two-dimensional wave tank using direct conversion by a dynamo PTO (Power Take-Off) system. The dynamic power-generation characteristics of the hybrid wave energy converter are analyzed with respect to the eventual regularity of the incident wave (regular and irregular wave conditions), and the data necessary for the design of the generator and control system are acquired.

**Keywords:** hybrid wave energy converter; horizontal cylinder; swing plate; dynamo PTO system; capture width ratio

## 1. Introduction

Huge investments are being poured into new renewable energies as part of the Green New Deal policies launched worldwide to solve at once the economic problems triggered by the pandemic as well as the climate crisis caused by unusual temperature events. Among the ocean renewable energy resources, wave power generation is attracting interest since it is able to secure clean energy from the ocean, which occupies two-third of the earth surface. However, the amount of energy that can be produced by wave power generation is difficult to predict accurately because of the large variability and high irregularity of the incident waves. In addition, there is a multitude of problems starting from the design to the offshore installation of the structure, including its safety. Compared with standardized offshore wind power energy, research on wave power generation is still under course considering the power generation method, the type of installation and the offshore conditions [1]. Korea has also undertaken an analysis of the wave distribution characteristics in the Jeju area using the SWAN model simulation according to the variation of the seasonal cycle, wave height and wind direction in order to assess the wave power generation method and applicability considering the offshore conditions [2].

Wave energy converters can be classified as attenuator, oscillating water column and overtopping device with respect to the underlying energy absorption principle. The attenuator harvests energy through the movement of an object reacting sensitively to the up-and-down and rotational motions of the incident wave [3–5]. The attenuator offers high conversion efficiency since it makes direct use of the wave energy but is structurally unsafe because of its exposure to the wave. The oscillating water column generates electricity through a special air turbine such as the Wells turbine or the impulse turbine activated by

the pumping of the air entrapped inside the chamber of the device caused by the variation of the water level inside the chamber following the periodic oscillation of the incident wave [6–8]. The oscillating water column offers safety and convenient maintenance of the structure but has relatively low energy conversion efficiency. Having a surface inclined in the direction of the incident wave and capturing the wave breaking into its storage reservoir, the overtopping device extracts energy by using the potential energy created by the difference between the mean sea level and the sea level [9–11]. The overtopping device presents a simple structure with outstanding safety but harvests electricity only beyond a definite water level.

The PTO (Power Take-Off) of wave energy can be classified into hydraulic motor systems [12,13], pneumatic air turbine transfer systems [14,15], hydro–turbine transfer systems [16,17], direct mechanical drive systems [18,19] and direct linear electrical drive systems [20,21]. Worldwide, 31 companies are actively working to exploit the direct mechanical drive system, whereas 21 companies have chosen the hydro–turbine transfer system as of 2020 [22].

The present study proposes a new hybrid wave energy converter composed of a horizontal cylinder and a swing plate to improve the wave energy conversion efficiency using both the potential energy (height) and kinetic energy (period) of the incident wave. The horizontal cylinder generates electrical energy by using the potential energy of the incident wave, whereas the swing plate produces electricity by using the kinetic energy of the water particles. Additional improvement is brought by additionally applying a direct mechanical drive system-based PTO converter. The power-generating performance of the proposed hybrid wave energy converter is evaluated experimentally through a hydraulic model test at a scale ratio of 0.3 in a two-dimensional wave tank using direct conversion by a dynamo PTO system. The dynamic power-generation characteristics of the hybrid wave energy converter are measured in real time and analyzed with respect to the eventual regularity of the incident wave, and the data necessary for the design of the generator and control system are acquired.

## 2. Hybrid Wave Energy Converter and Test Method

### 2.1. Hybrid Wave Energy Converter

As shown in Figure 1, the hybrid wave energy conversion device developed for nearshore at depths below 5 m is composed of a horizontal cylinder, a swing plate, a clutch gear and a yawing plate. The horizontal cylinder generates electricity by its rotation triggered by the water particle velocity and the current velocity. Moreover, electric power is also harvested by the heave motion caused by the height of the incident wave. The swing plate produces electricity by pendulum motion induced by the oscillation of the incident wave. Separate clutch gears are adopted for each horizontal cylinder and swing plate to minimize the energy loss of the incident wave and improve the capture width ratio of the wave energy converter. The clutch gears allow electricity to be generated by the horizontal cylinder when the rotation or heave motion velocity of the cylinder is higher than the swing velocity of the swing plate and by the swing plate when the opposite occurs. Thanks to such principle, the capture width ratio of the hybrid wave energy converter can be enhanced through the activation of the converter by instantaneously selecting the higher capture width ratio provided by the horizontal cylinder using the circular motion or kinetic motion of the incident wave or by the swing plate exploiting the linear oscillation. A period of suspension for the converter may occur when the water particle motion changes direction when the incident wave moves from wave crest to wave trough. However, the application of the clutch gears prevents such occurrence and enables the converter to rotate continuously in one direction, which in turn minimizes fundamentally any loss of the incident wave energy. The yawing plate floats, which makes it adaptable to water level change, and can rotate in both directions regardless of the ever-changing direction of the incident wave. The main supporting structure acting as the pivot of a seesaw and the

added mass located as counterpart of the horizontal cylinder enable to control smoothly and easily the draft of the horizontal cylinder converter.

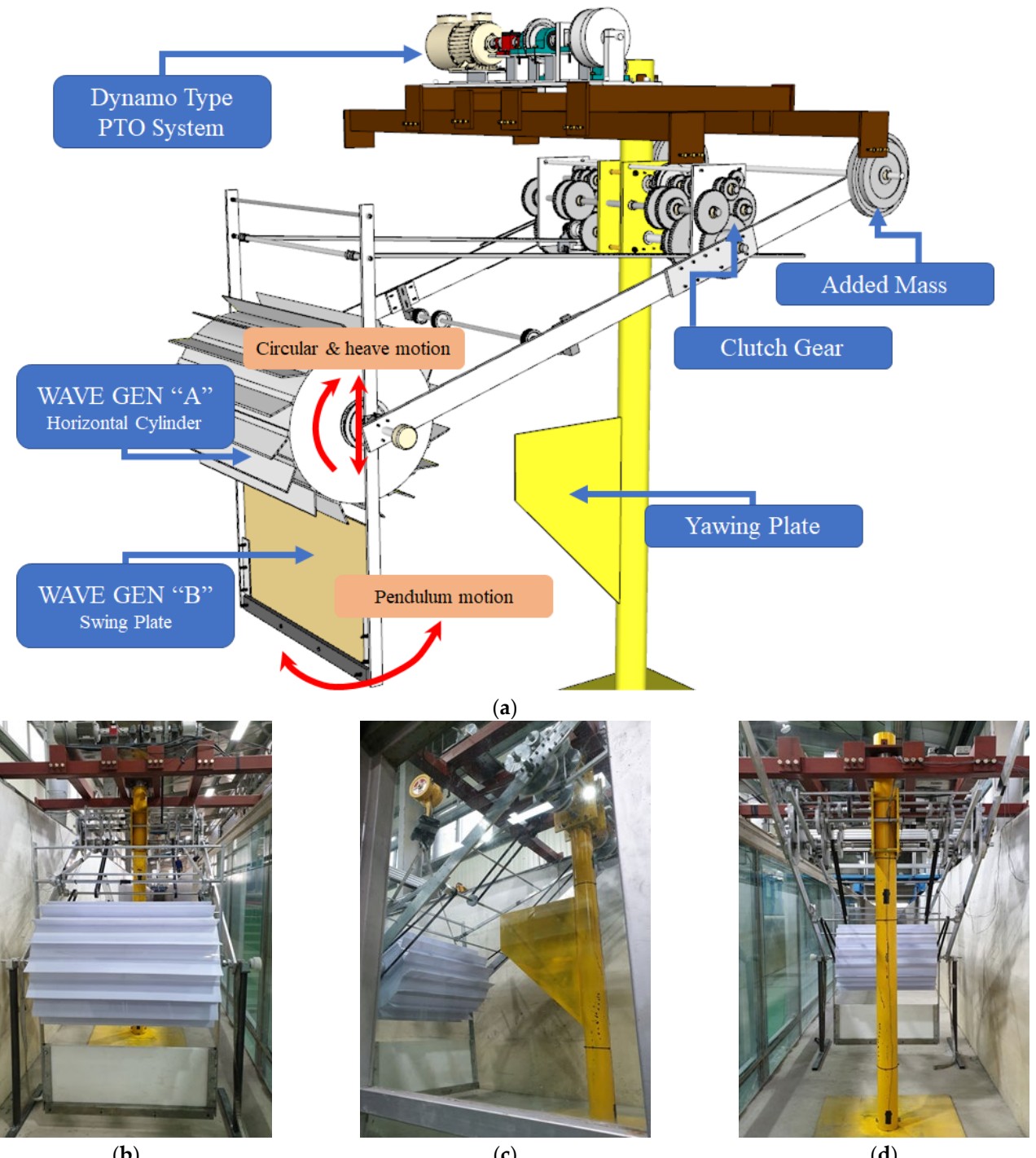

**Figure 1.** Unit module system of hybrid wave energy converter for nearshore: (**a**) design drawing and designation of components; (**b**) view from horizontal cylinder side; (**c**) side view; (**d**) view from added mass side.

## 2.2. Dynamo PTO System and Monitoring System

The output of the hybrid wave energy converter driven by the combined horizontal cylinder and swing plate is schematized in Figure 2. The developed converter applies the PTO system for wave energy generation exhibiting the characteristics of the dynamo proposed by Lee et al. [23]. The servo motor system controls the mechanical output of the

wave energy converter to adjust it to the predefined load torque. The resulting mechanical output is measured in real time by a monitoring system composed of a torque sensor and an encoder. The torque device adjusts stepwise the mechanical load of the wave energy converter by a 2.2 kW servo motor control that can manage constant torque. Figure 3 describes the composition of the dynamo PTO system and monitoring system. The power generated by the wave energy converter can be calculated as follows based upon the measured torque and angular velocity. The capture width ratio (CWR) of the system can be analyzed by comparing the generated power with the incident energy produced by the wave generator.

$$P = T \times \omega, \tag{1}$$

where $P$ = generated power; $T$ = torque and $\omega$ = angular velocity.

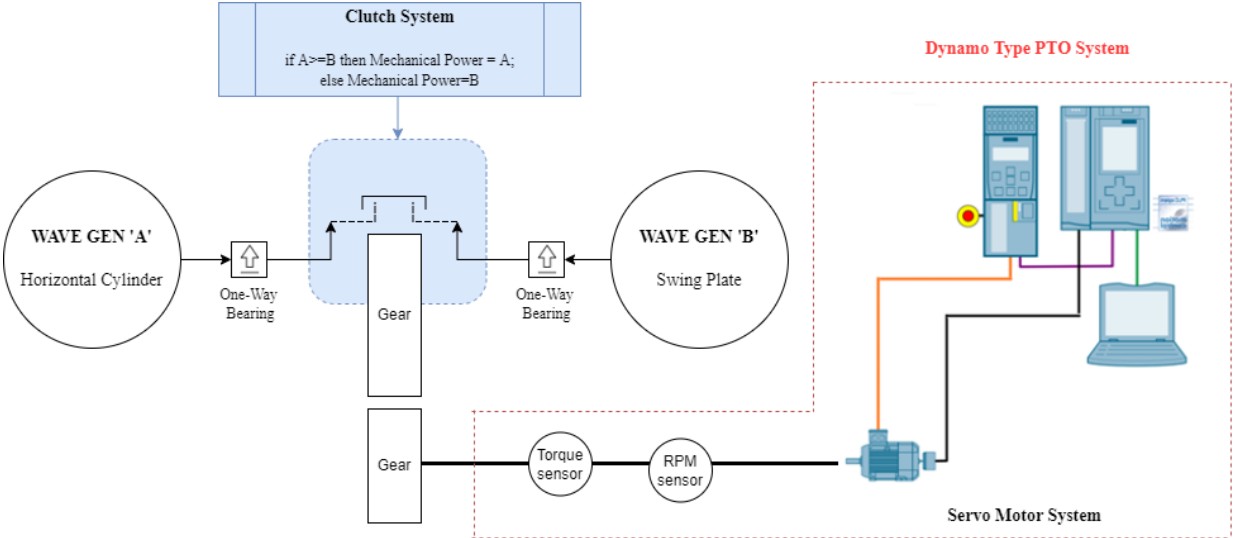

**Figure 2.** Experimental schematic diagram of dynamo type PTO system and monitoring system.

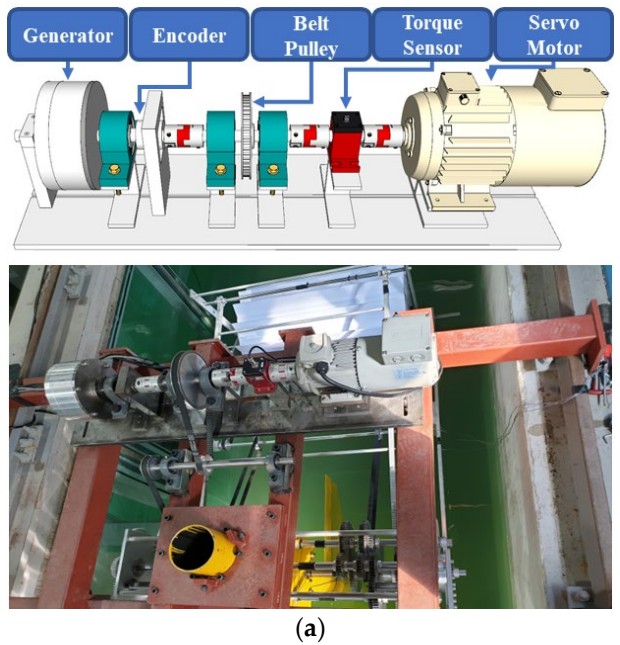
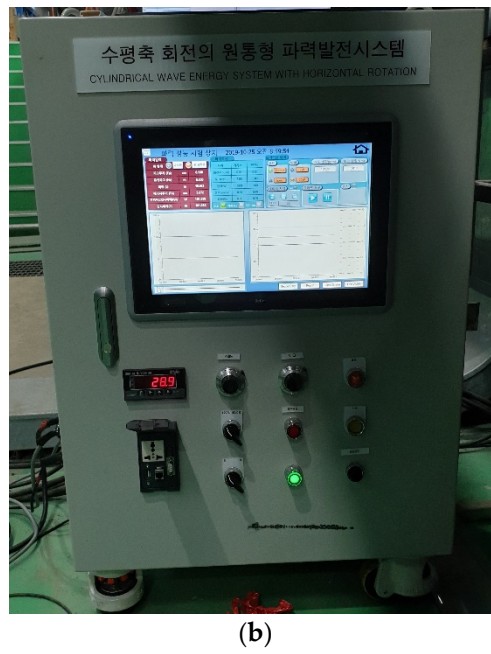

(**a**)  (**b**)

**Figure 3.** Set up of dynamo type PTO system and monitoring system: (**a**) composition of dynamo type PTO system; (**b**) control box of monitoring system.

The monitoring system adopted in the dynamo PTO system measures the characteristics of the electric power generated by the wave energy converter from the incident wave of the wave tank every 0.1 s, and the computed data can be consulted in real time. The GUI (Graphical User Interface) of the monitoring system is set to allow simultaneous comparison of the wave state, servo motor control status and generated power as well as to store data. The test conditions such as the period, height, length and depth of the incident wave are input, and their mean values can be checked. The control status of the servo motor involves the setting of the mechanical load torque, the on/off state of control and the communication status that can be checked in real time. The generated power can be computed using Equation (1) based upon the measured torque and rotation cycle data.

### 2.3. Two-Dimensional Wave Tank and Hydraulic Model Test Conditions

The hydraulic test was conducted in the two-dimensional wave–current–tide complex wave tank at the Experimental Center for Coastal and Harbor Engineering located in the Yeosu Campus of Chonnam National University. The wave tank presents a length of 100 m, a width of 2 m and a height of 3 m that can produce a maximum regular wave height of 130 cm, a wave period of 0.5 to 8.0 s and a maximum current velocity of 1.0 m/s. As shown in Figure 4, three capacity-type wave gauges were attached to the front of the column frame of the converter at 2.5 m and 5.5 m and in the rear at 3.0 m to measure the height of the incident wave produced by the wave generator. The hybrid wave energy converter was installed at a spot about 70 m from the wave generator. A water depth of 1.5 m in the tank was applied according to the law of similarity with a scale of 0.3. Table 1 arranges the detailed specifications of the fabricated reduced-scale prototype of the hybrid wave energy converter with a similitude of 0.3 that was installed in the wave tank.

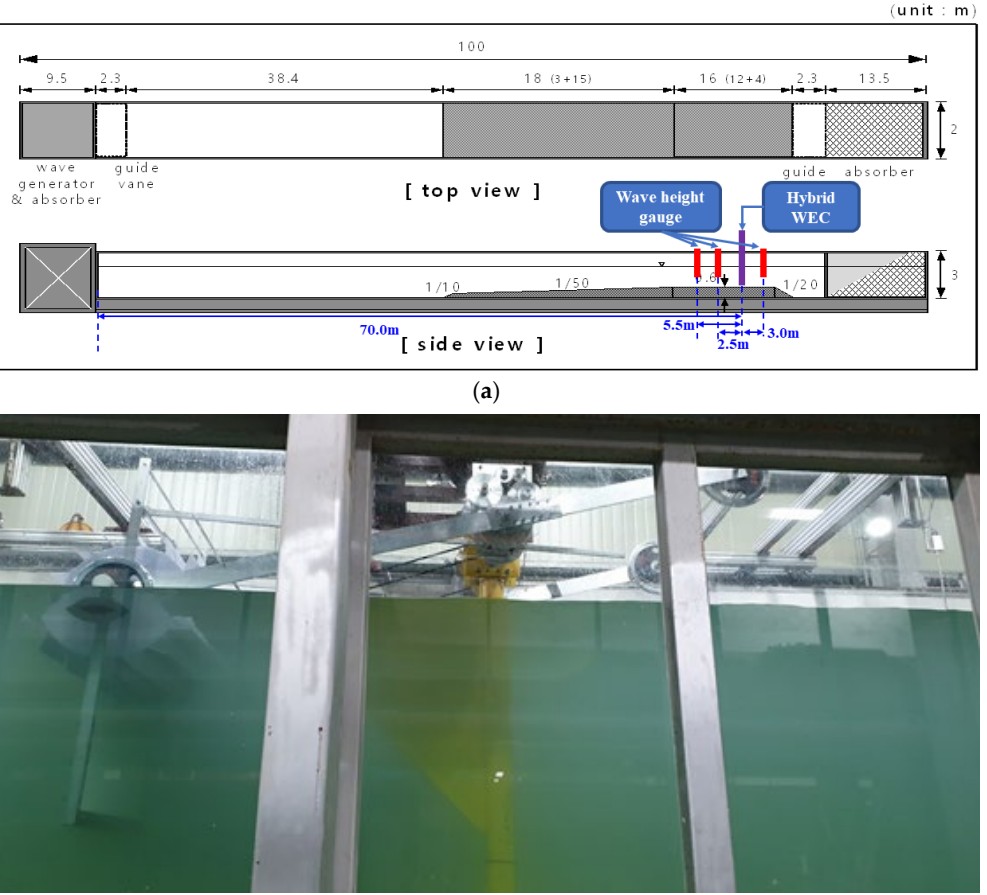

**Figure 4.** Two-dimensional wave tank: (**a**) definition sketch; (**b**) photograph of test.

**Table 1.** Detailed specifications of hybrid wave energy converter.

| Wave Gen A | Length (mm) | Diameter (mm) | Wing Length (mm) | No. of Wings | Weight (N) | Draft (mm) |
|---|---|---|---|---|---|---|
| Horizontal cylinder | 1000 (3333) | 600 (2000) | 90 (300) | 16 (16) | 467.071 (17,298.9) | 135 (450) |

| Wave Gen B | Length (mm) | | Height (mm) | | Thickness (mm) | | Weight (N) |
|---|---|---|---|---|---|---|---|
| Swing plate | 1000 (3333) | | 450 (1500) | | 20 (66.666) | | 568.786 (21,066.1) |

Values in parentheses indicate real values.

Tests were performed for six regular waves listed in Table 2 and six irregular waves using the PM spectrum (Pierson–Moskowitz spectrum) listed in Table 3 to consider various incident wave conditions. As shown in Figure 5, the tests were repeated 3 times for each wave period and height to compute the mean wave period and mean wave height. The mean incident energy flux for the regular waves is defined as follows [24].

$$P_{wre} = \frac{1}{8}\rho g H^2 c_g,$$ (2)

$$c_g = \frac{1}{2}\left(1 + \frac{2kh}{\sinh 2kh}\right)\frac{L}{T}$$

where $P_{wre}$ = energy flux of regular wave; $\rho$ = density of water; $g$ = gravitational acceleration; $H$ = incident wave height; $c_g$ = group velocity; $L$ = wave length; $k$ = wave number ($2\pi/L$); $h$ = water depth and $T$ = wave period.

**Table 2.** Regular wave conditions for experimental test.

| Designation | Wave Period, $T$ (s) | Wave Height, $H$ (m) | Wave Length, $L$ (m) | Wave Steepness, $H/L$ | Group Velocity, $c_g$ (m/s) | Energy Flux, $P_{wre}$ (W) |
|---|---|---|---|---|---|---|
| RE 1 | 2.742 (5.006) | 0.330 (1.099) | 9.105 (30.551) | 1/27.609 | 2.542 | 338.66 |
| RE 2 | 3.291 (6.009) | 0.282 (0.939) | 11.444 (38.147) | 1/40.640 | 2.884 | 280.19 |
| RE 3 | 3.846 (7.022) | 0.278 (0.927) | 13.743 (45.809) | 1/49.417 | 3.116 | 295.17 |
| RE 4 | 4.379 (7.995) | 0.311 (1.035) | 15.910 (53.034) | 1/51.224 | 3.269 | 386.31 |
| RE 5 | 4.930 (9.001) | 0.264 (0.881) | 18.123 (60.409) | 1/68.543 | 3.382 | 289.64 |
| RE 6 | 6.572 (11.999) | 0.286 (0.952) | 24.617 (82.058) | 1/86.164 | 3.575 | 357.42 |

Values in parentheses indicate real values.

**Table 3.** Irregular wave conditions with PM spectrum for experimental test.

| Designation | Zero-Crossing Wave Period, $T_2$ (s) | Energy Period, $T_e$ (s) | Significant Wave Height, $H_s$ (m) | Energy Flux, $P_{wir}$ (W) |
|---|---|---|---|---|
| IRRE 1 | 2.671 (4.877) | 3.232 (5.901) | 0.268 (0.892) | 79.9 |
| IRRE 2 | 2.987 (5.453) | 3.614 (6.599) | 0.258 (0.859) | 75.37 |
| IRRE 3 | 3.429 (6.260) | 4.149 (7.575) | 0.323 (1.076) | 119.86 |
| IRRE 4 | 4.089 (7.465) | 4.948 (9.033) | 0.319 (1.062) | 118.07 |
| IRRE 5 | 4.987 (9.105) | 6.034 (11.017) | 0.368 (1.227) | 158.37 |
| IRRE 6 | 7.376 (13.467) | 8.925 (16.295) | 0.317 (1.055) | 117.39 |

Values in parentheses indicate real values.

Irregular waves are represented as the superposition of regular waves. The mean energy flux of an irregular wave in shallow water can be expressed as follows:

$$P_{wir} \approx \frac{\rho g}{64\pi}\left[\tanh\left(\frac{4\pi^2 h}{g T_e^2}\right)^{3/4}\right]^{2/3} T_e H_s^2$$ (3)

where $P_{wri}$ = energy flux of irregular wave; $T_e$ = energy period ($0.86 \times T_p$); $T_p$ = peak wave period ($1.407 \times T_2$); $T_2$ = zero crossing wave period and $H_s$ = significant wave height.

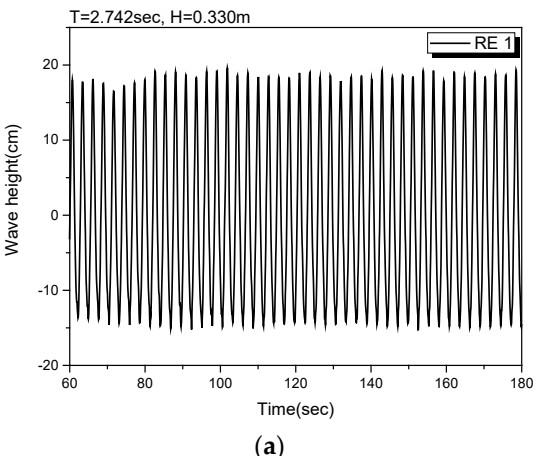

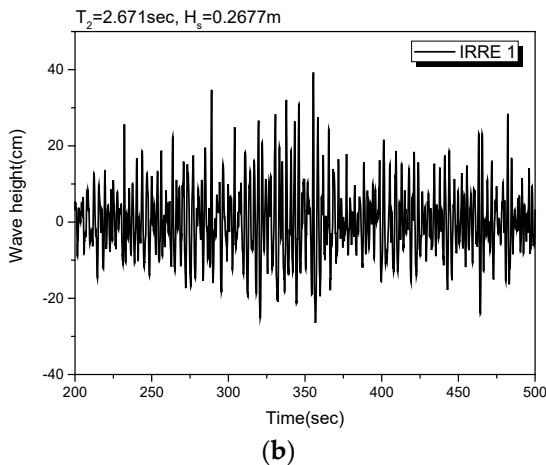

(**a**)

(**b**)

**Figure 5.** Measured wave height data: (**a**) regular wave case; (**b**) irregular wave case.

### 3. Experimental Results and Discussion

The number of rotations and the torque were measured in real time when the load torque for control was given in order to check the output and control status of the dynamo PTO system. These measurements were converted to calculate the generated power for comparison. Figure 6 plots the real-time characteristics of the generated power for regular wave case RE 1, which exhibited the largest variation of the number of rotations of the main axis and the highest generated power among the regular waves. Figure 6 shows the real-time output during five wave periods of the incident wave when the mechanical load torque of 0.0231 Nm was burdened by the servo motor. The power generated by a regular wave presents characteristics resulting from the combination of the outputs produced by both the horizontal cylinder and swing plate within the period (2.742 s) of the incident wave. The curve of the generated power exhibits repeatedly the continuity of the incident wave period. The power generated by the common attenuator, which converts directly the energy of the wave, displays zero twice within the incident wave period, but the output of the proposed hybrid wave energy converter approaches zero only once for the regular wave. The high power generated around 1/4 period and 3/4 period indicates that the swing plate of the hybrid wave energy converter generates maximum kinetic energy at wave crest and wave trough during one period of the regular wave. At 1/2 period, when the swing plate does not produce energy due to the change in the direction of its motion, the generated power exceeds 30 W owing to the action of the horizontal cylinder which exploits the circular motion and heave motion of the incident wave. This result demonstrates that the clutch gears adopted in this study enable to combine smoothly and continuously the electric powers generated by the horizontal cylinder and swing plate. The draft of the horizontal cylinder could be easily adjusted by a mechanism where the main supporting structure acts as the pivot of a seesaw with the horizontal cylinder at one end and an added mass at the other end. This seesaw did not affect the power generation when the incident wave turned from crest to trough but made the generated power close to zero when the incident wave turned from trough to crest. For the regular wave cases, the number of rotations (RPM) of the wave energy converter exhibited periodic continuity and stable power generation with very small variability in each period. The load torque remained practically constant, which advocates the possibility for stable power generation adapted to the incident wave characteristics by the proposed hybrid wave energy converter.

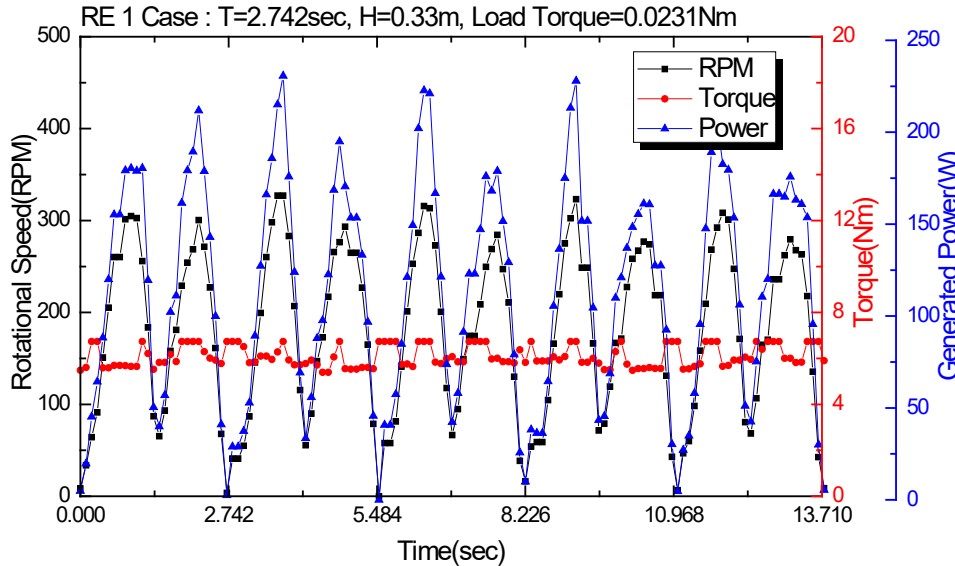

**Figure 6.** Real-time generated power, torque and rotational speed for RE 1 case.

A series of tests were conducted for nine load torques of the servo motor ranging between 0 and 0.385 Nm, namely 0.0 Nm, 0.0077 Nm, 0.0231 Nm, 0.0385 Nm, 0.0771 Nm, 0.1542 Nm, 0.2313 Nm, 0.3084 Nm and 0.3855 Nm, to evaluate the CWR of the hybrid wave energy converter for regular wave conditions. Figures 7 and 8 plot, respectively, the mean power and instant maximum power generated during 2 min under regular wave condition. Higher power appears to have been generated at load torques below 0.0771 Nm, whereas the generated power seems to experience a significant reduction beyond 0.0771 Nm. Moreover, the highest power generation occurred for regular wave case RE 1, with the shortest period and the generated power decreasing with longer periods of regular waves. It is noteworthy that the minimum power generation occurred for regular wave case RE 3. Considering that the proposed hybrid converter is purposed to be installed in a marine environment at depths no greater than 5 m, the wave conditions to be met will mostly correspond to regular wave cases RE 1 and RE 2. Consequently, the proposed hybrid wave energy converter appears to be the most suited for nearshore application, as targeted in its development.

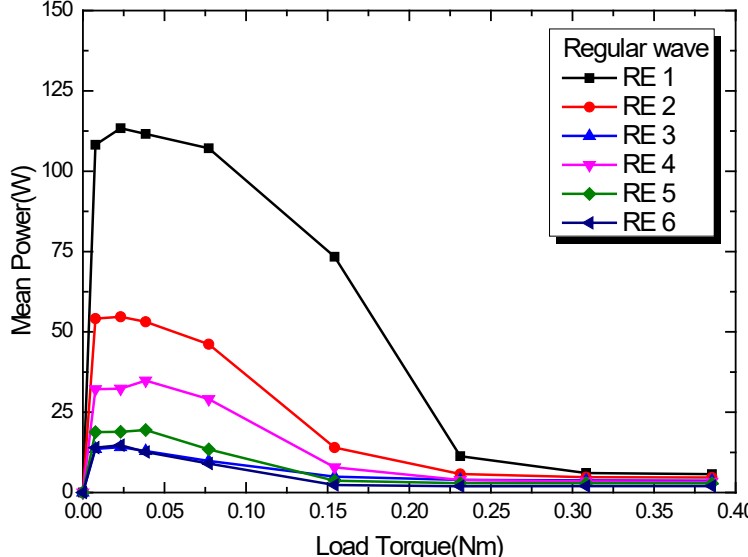

**Figure 7.** Mean power according to load torque for various regular wave conditions.

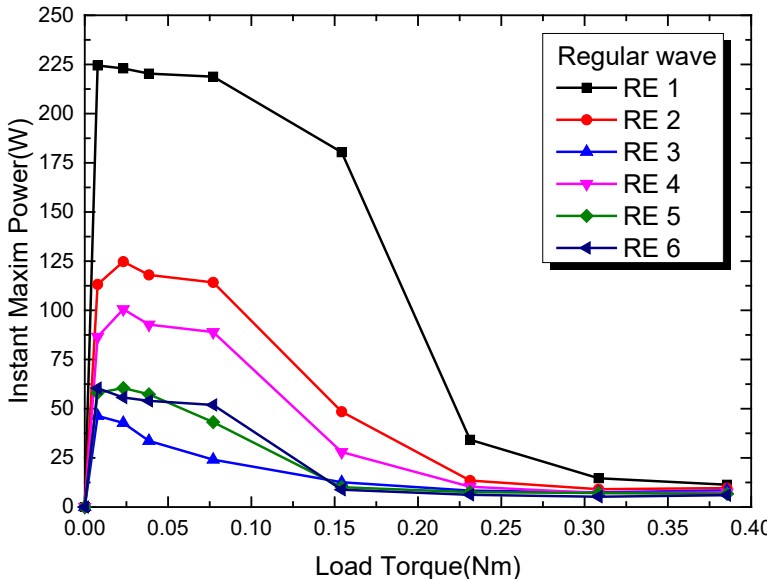

**Figure 8.** Instant maximum power according to load torque for various regular wave conditions.

Figure 9 plots the mean CWR of the hybrid wave energy converter according to the variation of the change in the load of the servo motor in regular wave conditions. As expressed in Equation (4), the CWR is the generated mean power measured during 2 min divided by the mean energy flux of the incident wave.

$$\text{CWR} = \frac{P(\text{generated mean power})}{P_w(\text{incident wave energy})} \tag{4}$$

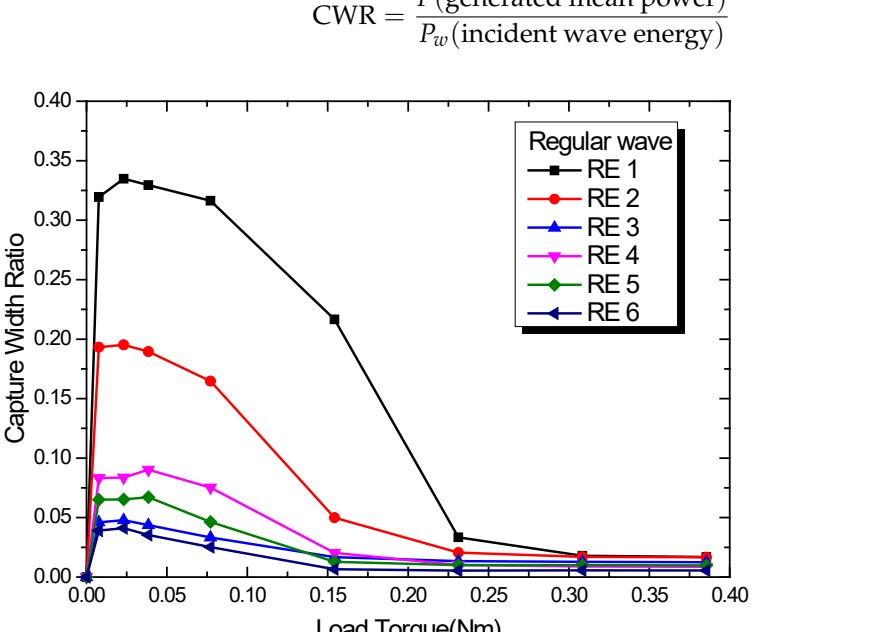

**Figure 9.** CWR of mean power according to load torque for various regular wave conditions.

For all the given regular wave conditions, the maximum CWR is developed for the load torque of the servo motor, ranging between 0.0077 Nm and 0.0385 Nm. The highest CWR (33.48%) is observed for regular wave case RE 1 with the shortest period, and the CWR tends to reduce with longer periods.

Table 4 arranges the maximum CWR of mean and instant maximum power for each of the considered regular wave conditions. It appears that the maximum instant CWR can convert 66.28% of the incident wave energy into electric power for regular wave case RE 1. A larger CWR is observed for shorter wave periods. The ratio of the instant maximum

CWR to the mean CWR ($P_{ma}/P_{me}$) ranges between 1.980 and 4.105, which indicates that the ratio tends to increase as much as the mean CWR decreases with longer periods of the incident wave.

**Table 4.** Maximum CWR of mean and instant maximum power for regular wave conditions.

| Wave Condition | Incident Wave Power, $P_{in}$ | Generated Mean Power, $P_{me}$ | Max. Mean CWR, $P_{me}/P_{in}$ | Instant Max. Power, $P_{ma}$ | Max. Instant CWR, $P_{ma}/P_{in}$ | $P_{ma}/P_{me}$ |
|---|---|---|---|---|---|---|
| RE 1 | 338.66 | 113.39 | 0.3348 | 224.46 | 0.6628 | 1.98 |
| RE 2 | 280.19 | 54.69 | 0.1952 | 124.74 | 0.4452 | 2.281 |
| RE 3 | 295.17 | 14.13 | 0.0479 | 46.35 | 0.157 | 3.28 |
| RE 4 | 386.31 | 34.85 | 0.0902 | 100.57 | 0.2603 | 2.886 |
| RE 5 | 289.64 | 19.46 | 0.0672 | 60.51 | 0.2089 | 3.109 |
| RE 6 | 357.42 | 14.72 | 0.0412 | 60.43 | 0.1691 | 4.105 |

Figure 10 plots the real-time power generation characteristics for irregular wave case IRRE 1 with the largest variation in RPM and the highest generated power among the irregular wave conditions using PM spectrum. Figure 10 presents the real-time output measured during 5 min resulting from the incident irregular wave generated by attributing a mechanical load torque of 0.0231 Nm to the servo motor. The proposed hybrid wave energy converter is able to generate power by exhibiting the characteristics shown in Figure 6 that fit in real time with the characteristics of the incident wave since the irregular wave is the result of the combination of various regular waves. In addition, the RPM of the wave energy converter shows large variation according to the incident wave condition throughout the irregular wave tests, whereas the torque seems to have small variation. Consequently, it appears that the hybrid wave energy converter reacts very sensitively to the change in the incident wave height.

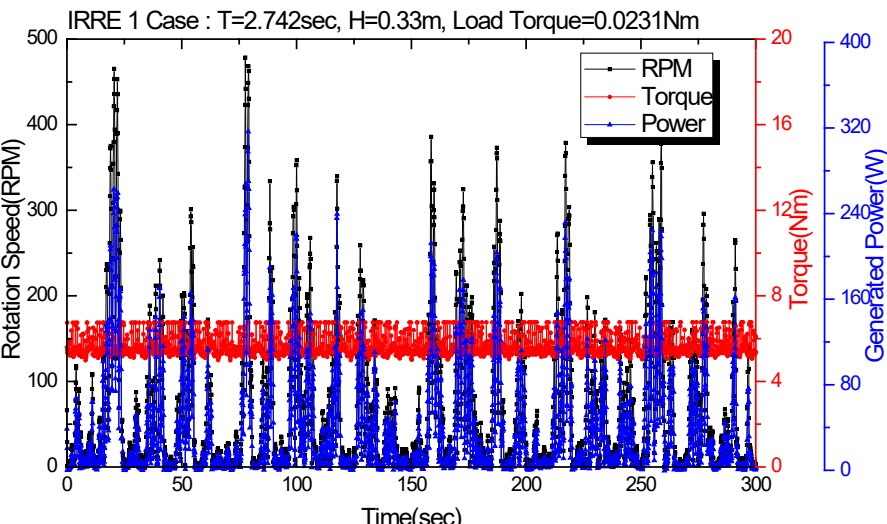

**Figure 10.** Real-time generated power, torque and rotational speed for IRRE 1 case.

A series of tests were conducted for nine load torques of the servo motor, namely 0.0 Nm, 0.0077 Nm, 0.0231 Nm, 0.0385 Nm, 0.0771 Nm, 0.1542 Nm, 0.2313 Nm, 0.3084 Nm and 0.3855 Nm, to evaluate the CWR of the hybrid wave energy converter for irregular wave conditions. Figures 11 and 12 plot, respectively, the mean power and instant maximum power generated during 5 min under irregular wave condition. Higher power appears to have been generated at load torques below 0.0771 Nm, whereas the generated power seems to experience gradual reduction beyond 0.0771 Nm. It is noteworthy that the mean generated power at load torque greater than 0.15 Nm exhibits very large values in irregular wave conditions differently from that in regular wave conditions. The mean generated

power does not experience sharp decreases even when the period of the incident wave becomes longer. Accordingly, the proposed hybrid wave energy converter can provide very stable power generation under irregular wave conditions similar to that occurring actually in a marine environment. In Figure 12, the values of the instant maximum generated power show irregularity unlike the similar mean power curves observed in regular wave conditions. This last observation indicates that the power generation reflects the irregular characteristic of the incident wave.

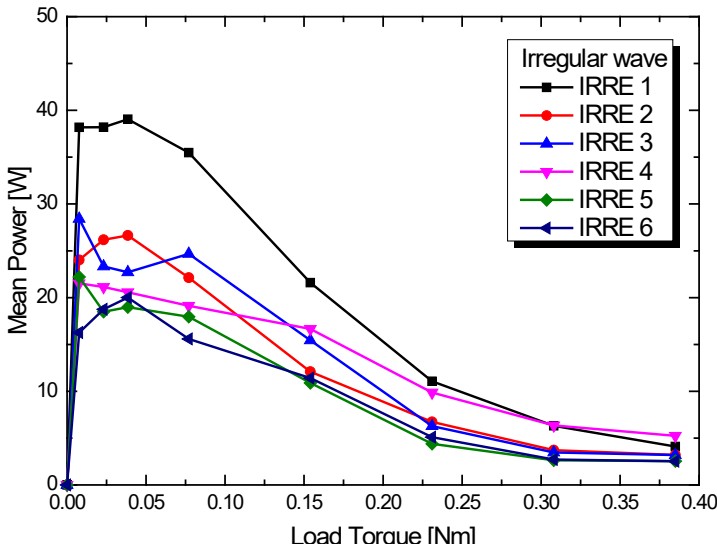

**Figure 11.** Mean power according to load torque for various irregular wave conditions.

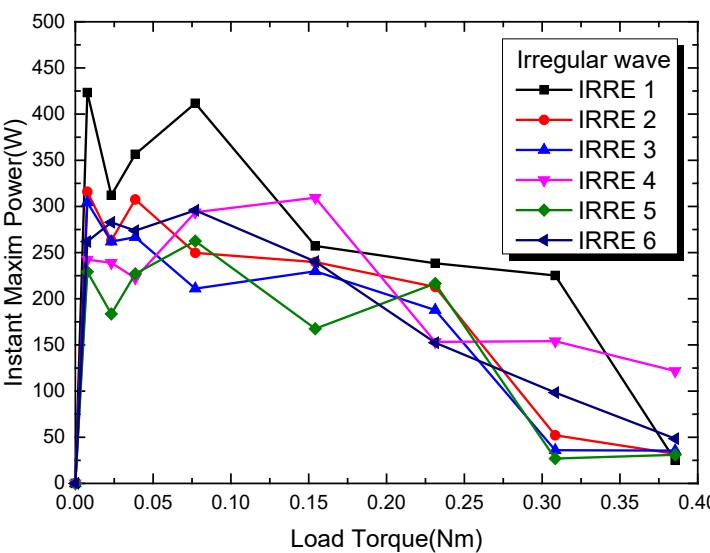

**Figure 12.** Maximum power according to load torque for various irregular wave conditions.

Figure 13 plots the mean CWR of the hybrid wave energy converter according to the variation of the change in the load of the servo motor in irregular wave conditions. The CWR is expressed as the generated mean power measured during 5 min divided by the mean energy flux of the incident wave. For the considered irregular wave conditions, the maximum CWR occurs when the load torque of the servo motor ranges between 0.0077 Nm and 0.0385 Nm, and the CWR tends to reduce proportionally to the increase in the applied load torque. Moreover, the largest CWR (48.87%) is observed for irregular wave case IRRE 1 with a short period, which represents an improvement by about 1.46 times compared with the regular wave condition (33.48%). It appears also that the reduction ratio of the

CWR with longer periods is smaller for the irregular wave condition than for the regular wave condition.

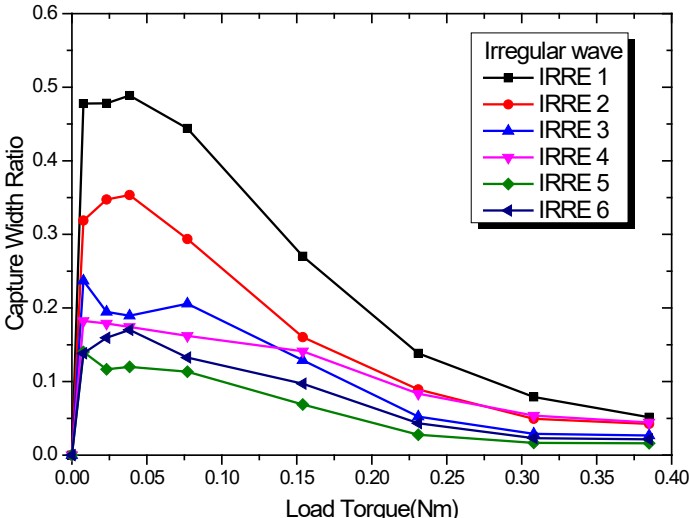

**Figure 13.** CWR of mean power according to load torque for various irregular wave conditions.

Table 5 arranges the maximum CWR of the mean and instant maximum power for each of the considered irregular wave conditions. The ratio of the instant maximum CWR to the mean CWR ($P_{ma}/P_{me}$) ranges between 1.980 and 4.105. Considering that the ratio of the mean height to the maximum height of a common incident irregular wave ranges between 2.56 and 3.2 ($H_{\max} = (1.6 \sim 2.0)H_{1/3} = (1.6 \sim 2.0) \times 1.6\overline{H} = (2.56 \sim 3.2)\overline{H}$), it appears that the proposed hybrid wave energy converter reacts very sensitively to the change in the incident wave height.

**Table 5.** Maximum CWR of mean and instant maximum power for irregular wave conditions.

| Wave Condition | Incident Wave Power, $P_{in}$ | Generated Mean Power, $P_{me}$ | Max. Mean CWR, $P_{me}/P_{in}$ | Instant Max. Power, $P_{ma}$ | Max. Instant CWR, $P_{ma}/P_{in}$ | $P_{ma}/P_{me}$ |
|---|---|---|---|---|---|---|
| IRRE 1 | 79.9 | 39.05 | 0.4887 | 365.5 | 4.5745 | 9.36 |
| IRRE 2 | 75.37 | 26.64 | 0.3535 | 307.6 | 4.0812 | 11.547 |
| IRRE 3 | 119.86 | 28.41 | 0.1796 | 304.4 | 2.5396 | 14.138 |
| IRRE 4 | 118.07 | 21.53 | 0.188 | 242.4 | 2.053 | 10.919 |
| IRRE 5 | 158.07 | 22.2 | 0.1402 | 229.2 | 1.4472 | 10.324 |
| IRRE 6 | 117.39 | 20.01 | 0.1705 | 273.7 | 2.3315 | 13.678 |

Figure 14 compares the real time power, mean incident energy and mean generated power measured during 5 min for irregular wave case IRRE 1 that exhibited the highest power generation among the tests. The real-time measured power is sorted in descending order to verify the frequency and level of the output. In view of the distribution of the sorted values computed at time steps of 0.1 s, the generated power exceeding the energy of the incident wave represents approximately 16.91% of the whole set, and the generated power exceeding the mean power represents about 32.43% of the whole set. This indicates that the clutch gears adopted to improve the CWR of the proposed hybrid wave energy converter are very efficient in irregular wave conditions. The CWR of previous wave energy converters provided limited mean power generation or were within a narrow load range. Additionally, by applying a dynamo PTO system, this study could propose wide-ranging mean power and instant maximum power by the hybrid wave energy converter and acquire diversified data necessary for the design of the wave energy converter.

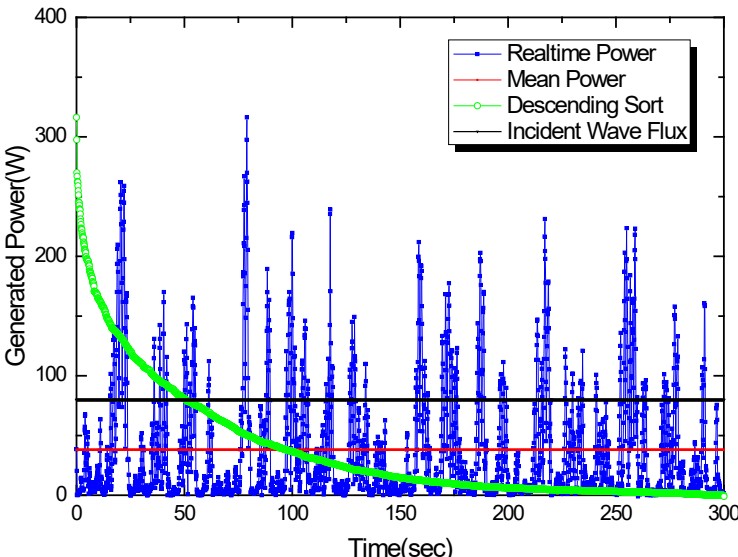

**Figure 14.** Real-time generated power and distribution of IRRE 1 case.

## 4. Conclusions

The power-generating performance of a hybrid wave energy converter was evaluated experimentally through hydraulic model testing at a scale ratio of 0.3 in a two-dimensional wave tank using direct conversion by a dynamo PTO system. The dynamic power-generation characteristics of the hybrid wave energy converter were measured in real time and analyzed for regular and irregular wave conditions of the incident wave. The following conclusions can be drawn:

(1) The horizontal cylinder and the swing plate applied in the hybrid wave energy converter could perform stable conversion of the potential energy (wave height) and cyclic kinetic energy (period) of the incident wave into electric energy.

(2) The clutch gears enabling to combine efficiently and continuously the two powers generated by the horizontal cylinder and the swing plate were very effective in improving the CWR of the hybrid wave energy converter.

(3) Compared with the regular wave conditions, the hybrid wave energy converter exhibited a CWR improved by about 1.46 times in irregular wave conditions and without sharp reduction in the generated mean power, even for a longer period of the incident wave. This indicated the potential of the hybrid wave energy converter for very stable power generation in a real marine environment.

(4) Considering that most wave conditions occurring in a marine environment at depths smaller than 5 m would present short periods, the proposed hybrid wave energy converter appeared to be very suitable for nearshore installation.

(5) The characteristics of the real-time generated power and the distribution of the power generation acquired through the dynamo PTO system provided precious data, such as the number of rotations, the torque and the electric voltage and current necessary for the design of generators and converters optimized for wave energy generation systems.

**Author Contributions:** Conceptualization and methodology, M.-S.P.; PTO system, S.-H.L. and S.-C.K.; experimental tests, M.-S.P. and S.-H.L.; validation and formal analysis, M.-S.P. and S.-C.K.; data curation, S.-H.L. and S.-C.K.; writing—original draft preparation, M.-S.P. and S.-H.L.; writing—review and editing, M.-S.P.; supervision, M.-S.P.; project administration, M.-S.P.; funding acquisition, M.-S.P. All authors have read and agreed to the published version of the manuscript.

**Funding:** Research for this paper was carried out under the KICT Research Program (project no. 20190493-001, Experimental test of wave energy system for shallow water) funded by the Ministry of Science and ICT.

**Acknowledgments:** This study was supported by the Yeosu Gwangyang Port Authority of South Korea, Project No: 20210772-001 (Planning research for application of mobile wave energy converter for upcycling of used batteries).

**Conflicts of Interest:** The authors declare no conflict of interest.

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
