# Peer review of "Experimental Capture Width Ratio on Unit Module System of Hybrid Wave Energy Converter for Nearshore"

_applsci, doi:10.3390/app12125845_

Round 1
Reviewer 1 Report
This paper designs a new wave energy converter that can utilise either the potential energy or the kinetic energy of waves depending on their relative intensity. Then a laboratory experimental is conducted to evaluate its performance, yielding convincing results. Overall, this is a good study, well organised, well written, and of practical significance. I would recommend its acceptance as long as my following concerns can be addressed.
- The reader would be happy to see a hydrodynamic study under irregular wave conditions because it is close to reality. So, the authors are suggested to highlight the irregular wave conditions in the abstract.
- Page 2, Lines 59-61. "31 companies are...whereas 21 companies have..." come from the reference [22] that was published on 2020. Owing to the rapid development of ocean energy, up to now the numbers must have increased. So, I would suggest the authors to provide the deadline for those statistical data.
- In Figure 1(a), the motion patterns of "WAVE GEN A" and "WAVE GEN B" should be depicted. It took me a while to understand how the swing plate moves like a pendulum.
- "kgf" in Table 1 is not an SI unit. Please change it to "N".
- Figure 5 is not cited in the main text. Please remove it or supplement some descriptions.
- Page 9, Lines 214-216. It is not clear that, at a water depth no greater than 5 m, why the regular wave conditions to be met will only be RE1 and RE2. How about RE4? It also produces a considerable mean power.
- Will the sediment movement on the sea bottom influence the operation of the new hybrid wave energy converter?
Reviewer 2 Report
The present manuscript presents a study of a new hybrid wave energy converter composed of a horizontal cylinder and a swing plate, which harvests both the potential energy of the incident wave and the kinetic energy of the water particles. The performance of the equipment were measured in a laboratory experiment. The manuscript is well-written and the proposed equipment shows great application prospects. However, the quality of the study would be further improved if a performance comparison with some other types of converter were made. At least, a discussion of the pros and cons of the current converter in comparison with others would be much appreciated.
Reviewer 3 Report
Min Su Park et al
This manuscript describes the mechanical and electrical performance of a hybrid wave power generator consisting of a rotating cylinder and a rigid plate, as determined in an experimental wave tank.
The concept of using the two devices to access different aspects of the power available in waves is good, and seems to work, although the fall off in mean power extracted at higher torques is not understood.
My main concern about the manuscript is the degree to which the results will be transferable to the real world. The power train and control systems appear to be mounted on a frame above the water, and that this frame is rigidly attached to the walls of the tank. I have concerns that the weather and waterproofing of the above water electrical systems have not been addressed. The authors state that the machine would be used in less than 5m depth of water. Such near shore locations can experience violent, large and breaking waves, and I am not sure that the suthors have considered such problems.
I also have concerns that in the real work the mounting for the power train and control systems will not be rigid, but will also be affected by the wave motion. This adds considerably to th degrees of freedom of movement of the entire device, which has not be simulated in the tank, and which may greatly alter the performance and survivability of the device.
Despite these concerns, and the need for a thorough linguistic editing, the concept is sufficiently interesting as to merit publication.
Round 2
Reviewer 1 Report
The manuscript can be accepted as it is.
Reviewer 2 Report
I suggest the manuscript be accepted for publication.